# Differential guest location by host dynamics enhances propylene/propane separation in a metal-organic framework

Dmytro Antypov [1], Aleksander Shkurenko [2], Prashant M. Bhatt [2], Youssef Belmabkhout[2], Karim Adil[2], Amandine Cadiau[2], Mikhail Suyetin[2], Mohamed Eddaoudi [2], Matthew J. Rosseinsky[1] & Matthew S. Dyer [1]✉

Energy-efficient approaches to propylene/propane separation such as molecular sieving are of considerable importance for the petrochemical industry. The metal organic framework **NbOFFIVE**-1-Ni adsorbs propylene but not propane at room temperature and atmospheric pressure, whereas the isostructural **SIFSIX**-3-Ni does not exclude propane under the same conditions. The static dimensions of the pore openings of both materials are too small to admit either guest, signalling the importance of host dynamics for guest entrance to and transport through the channels. We use ab initio calculations together with crystallographic and adsorption data to show that the dynamics of the two framework-forming units, polyatomic anions and pyrazines, govern both diffusion and separation. The guest diffusion occurs by opening of the flexible window formed by four pyrazines. In **NbOFFIVE**-1-Ni, $(NbOF_5)^{2-}$ anion reorientation locates propane away from the window, which enhances propylene/propane separation.

[1] Department of Chemistry, University of Liverpool, Crown Street, Liverpool L69 7ZD, UK. [2] King Abdullah Universitrey of Science and Technology (KAUST), Physical Sciences and Engineering Division, AMPM Center, Functional Materials Design, Discovery and Development Research Group (FMD3), Thuwal, Saudi Arabia. ✉email: msd30@liverpool.ac.uk

Propylene is an olefin raw material for petrochemical production, second only in importance to ethylene, and an essential building block for polypropylene, one of the most versatile thermoplastic polymers. As polypropylene synthesis requires high-purity propylene (99.5 weight %), removing other hydrocarbons, particularly propane, from propylene is one of the critical separations in the modern chemical industry. This is currently accomplished by energy-intensive cryogenic distillation[1], based on the small boiling point difference between the two compounds. Separation methods based on porous materials that utilise the difference between molecular sizes and diffusivities[2–5] or attractive host–guest interactions[6,7] are of great interest as energy-efficient alternatives.

Metal-organic frameworks (MOFs) are particularly attractive for propylene/propane separation as their modular crystalline structure allows selection of inorganic and organic building units to fine-tune pore size and internal surface functionality[8]. The recently reported **NbOFFIVE**-1-Ni, (as-made composition: [Ni(NbOF$_5$)(C$_4$H$_4$N$_2$)$_2$·2H$_2$O]), also referred to as KAUST-7[9], exhibits excellent separation performance. In **NbOFFIVE**-1-Ni, each Ni$^{2+}$ cation is coordinated to two (NbOF$_5$)$^{2-}$ polyatomic anions and four pyrazine molecules to form a primitive tetragonal lattice with **pcu** topology—an array of parallel inorganic pillars interconnected by pyrazine linkers that form parallel square-grid layers (Fig. 1a) with associated narrow one-dimensional channels that define the porosity of the material. There are a number of similar materials built from pyrazine molecules, metal cations, and polyatomic anions that have the same topology as **NbOFFIVE**-1-Ni, most notably the **SIFSIX**-3-M family where (SiF$_6$)$^{2-}$ anions

coordinate divalent metal cations (M = Cu, Zn, Ni, Fe)[10,11]. These materials precisely position a single guest molecule in their pore cavity and show increased affinity to carbon dioxide[12] when the metal is changed from Zn$^{2+}$ to Cu$^{2+}$ and to propyne[13] when the metal is changed from Zn$^{2+}$ to Ni$^{2+}$.

**NbOFFIVE**-1-Ni and **SIFSIX**-3-Ni differ only by the spacer anions that interconnect the Ni-pyrazine square-grid layers. **NbOFFIVE**-1-Ni was reported to rapidly adsorb propylene (58 mg/g at 298 K and 1 bar) but, unlike **SIFSIX**-3-Ni, the amount of propane adsorbed over several hours under the same condition was negligible[9].

In this work, we show how the host structure changes in the presence of propylene and propane guests and identify the specific molecular mechanisms present in **NbOFFIVE**-1-Ni (but absent in **SIFSIX**-3-Ni) that lead to its exceptional propylene/propane selectivity.

## Results

**NbOFFIVE-1-Ni structure and experimental gas adsorption data.** We report the structure of desolvated **NbOFFIVE**-1-Ni (Supplementary Table 1) that shows two distinct anion orientations (Fig. 1b, Supplementary Fig. 1) and an ordering of pyrazine tilts differing from that in the as-made material, [Ni(NbOF$_5$)(C$_4$H$_4$N$_2$)$_2$·2H$_2$O]. The C$_4$H$_4$N$_2$ pyrazine linkers and (NbOF$_5$)$^{2-}$ polyatomic anions in **NbOFFIVE**-1-Ni (used henceforth to refer to the desolvated material) form a network of intra-framework C–H···F contacts in which each anion interacts with eight pyrazines via its four equatorial fluorides (Fig. 1c) and each pyrazine

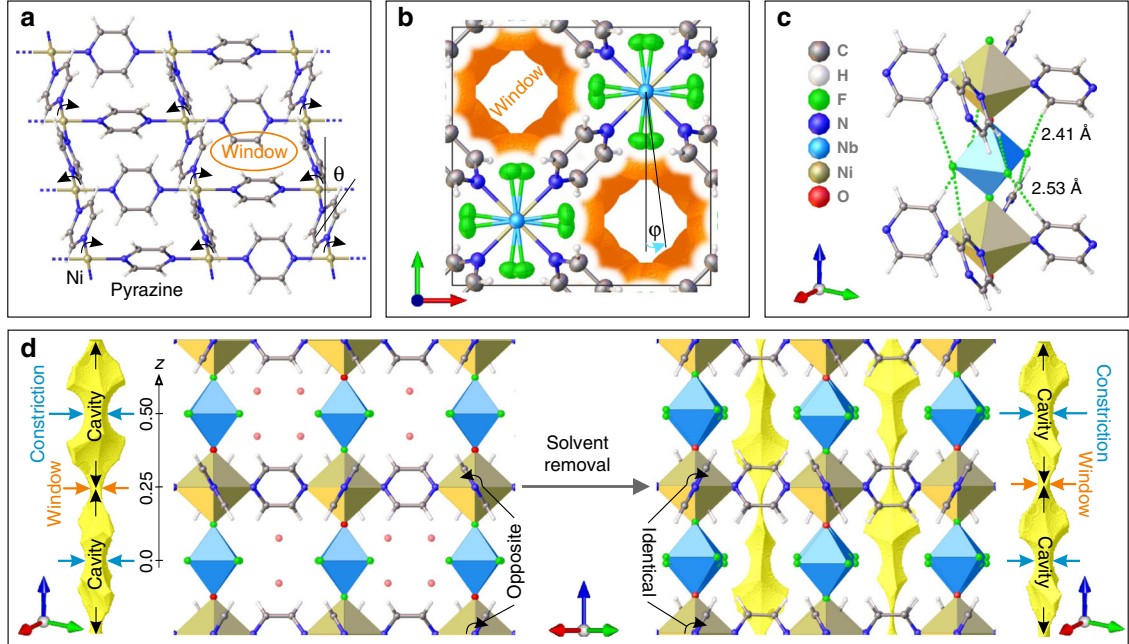

**Fig. 1 Structure of [Ni(NbOF$_5$)(C$_4$H$_4$N$_2$)$_2$]. a** The structure of the square-grid Ni-pyrazine layer with windows formed by four adjacent pyrazines. The arrows show the alternating tilts of pyrazine molecules within this layer, defined by $\theta$, observed in all structures studied. **b** The view along the $c$ axis in the desolvated material showing a unit cell containing two parallel one-dimensional channels (displacement ellipsoids corresponding to 50% probability are shown for all non-hydrogen atoms, the orange surfaces are van der Waals surfaces, the blue arrow indicates the rotation of the (NbOF$_5$)$^{2-}$ anion by $\varphi =$ 7.5° owing to the unequal interactions with the adjacent pyrazine layers). **c** An octahedral (NbOF$_5$)$^{2-}$ anion forms two types of C–H···F contacts (green dotted lines, distances shown) with two adjacent Ni-pyrazine layers: shorter contacts with the top layer and longer contacts with the bottom layer. Each equatorial fluoride forms two contacts with pyrazines that are not equivalent. **d** The structure of as-made material [Ni(NbOF$_5$)(C$_4$H$_4$N$_2$)$_2$·2H$_2$O] with the two water molecules in each cavity disordered over four equivalent sites (a larger I4/mcm unit cell left, the fractional coordinate $z$ used throughout the paper to describe guest location is shown) and desolvated material (a smaller P4/nbm unit cell right). The opposite tilts ($\theta = 30.5°$) of pyrazine linkers in neighbouring Ni-pyrazine layers in the as-made solvated material change to identical tilts ($\theta = 27.5°$) in the desolvated material. This changes the shape of the channels, shown in yellow for both structures as the volume accessible by the centre of a probe 2.4 Å in diameter. The channel consists of interconnected cavities separated by windows, with the constriction at the cavity centre defined by the polyatomic anions.

interacts with four anions through its four C–H bonds. The difference in the axial Nb–F and Nb–O bond lengths, respectively, 2.020(5) Å and 1.831(4) Å, of an $(NbOF_5)^{2-}$ anion results in equatorial fluorides being closer to one of the two neighbouring Ni-pyrazine layers. This produces shorter and stronger C–H⋯F interactions with this layer (Fig. 1c) and results in anion rotation by $\varphi = 7.5(1)°$ around the $c$ axis (Fig. 1b), consistent with density functional theory (DFT) calculations (Supplementary Fig. 2). The dynamic nature of the second building block of **NbOFFIVE**-1-Ni, the pyrazines, is demonstrated by the single crystal to single-crystal transition upon water removal, where the as-made material changes its symmetry from as-made I4/mcm to P4/nbm— while the pyrazine tilts remain strictly alternating within each Ni-pyrazine layer as shown in Fig. 1a, the orientation of equivalent pyrazine molecules in the adjacent layers changes from opposite to identical (Fig. 1d). This reorientation changes the shape of the one-dimensional channel but does not affect the position of the narrowest part of the channel, the window, that is defined by four pyrazines in the Ni-pyrazine layer via the Ni-pyrazine-Ni distance, nor that of the constriction at the centre of the cavity located close to the polyatomic $(NbOF_5)^{2-}$ anions, which is the next-narrowest point (Fig. 1d). The minimum dimensions of the channel are thus set by the two distinct framework-forming ligands. The polyatomic $(NbOF_5)^{2-}$ anion defines the separation between the windows and thus the length of the cavity through the Ni-NbOF5-Ni contact of 7.8311(4) Å. The window (2.724(8) Å) and constriction (3.22(1) Å) diameters of desolvated **NbOFFIVE**-1-Ni are much smaller than the kinetic diameter of either propylene (4.0 Å) or propane (4.3 Å), so the transport of these guests is expected to require the dynamical rotation of both the pyrazine molecules and the anions away from their equilibrium orientations.

Both desolvated **SIFSIX**-3-Ni (ref. [9]) and **NbOFFIVE**-1-Ni (this work) have similar Ni-Ni distances of, respectively, 7.0012 (2) Å and 7.0209(3) Å within the Ni-pyrazine layers but have different cavity lengths: the distance between the layers is 7.5655 (7) Å in **SIFSIX**-3-Ni and 7.8311(4) Å in **NbOFFIVE**-1-Ni. As $(SiF_6)^{2-}$ is smaller than $(NbOF_5)^{2-}$ (Si–F 1.639(2) Å, Nb–F 1.917 (5) Å), the constriction in **SIFSIX**-3-Ni (3.683(5) Å) is wider than that in **NbOFFIVE**-1-Ni (3.22(1) Å). Unlike **NbOFFIVE**-1-Ni, which at 25 °C and 1 bar adsorbs less than 3 mg/g of propane compared with 58 mg/g of propylene (Supplementary Fig. 3a), **SIFSIX**-3-Ni adsorbs both propane (45 mg/g) and propylene (76 mg/g) under the same conditions (Supplementary Fig. 3b). Propane uptake in **NbOFFIVE**-1-Ni increases as the pressure is increased and at 7 bar reaches, respectively, 47 mg/g and 42 mg/g at 35 °C and 50 °C. The open adsorption isotherms (Supplementary Fig. 4) show that the adsorption and desorption branches do not reach equilibrium at either 35 °C or 50 °C. In the competitive adsorption of equimolar propylene/propane mixture at 25 °C the co-adsorption of propane is expected to be very low as the total uptake of the mixture closely matches the uptake of pure propylene up to 3 bar at the same partial pressure of propylene (Supplementary Fig. 5a) as if propane was not present in the mixture. The gas chromatography (GC) analysis of the mixture uptake at 1.3 bar (Supplementary Fig. 5b) confirms that over 99% of adsorbed mixture is propylene, whereas the amount of co-adsorbed propane is below the experimental error of 2%.

**DFT calculations of propylene and propane in NbOFFIVE-1-Ni.** To investigate dynamic changes to the host associated with guest transport through the narrowest part of the one-dimensional channel of **NbOFFIVE**-1-Ni, we used DFT to identify adsorption sites near the window and to build the minimum energy path (MEP) associated with guest transport

from one side of the window to the other. In the case of propylene, all calculations converged to the same site (shown by five overlapping empty blue circles on each side of the window in Fig. 2), signalling directed guest–host interactions, whereas several adsorption configurations with similar energies were observed for propane (shown by five overlapping empty red squares on each side of the window in Fig. 2), signalling a rugged energy landscape. The MEP connecting the opposite sides of the window was then determined using the DFT Nudged Elastic Band (NEB) method[14].

Figure 2 shows the snapshots and the relative energies of the adsorption sites near the window as a function of the position of the central carbon on the guest. There are two neighbouring cavities centred at $z = 0$ and $z = 0.5$ separated by a window at $z = 0.25$ (SI methods). There are well-defined adsorption sites **A** and **B′** for propylene and broad adsorption sites **C** and **D′** for propane (primes indicate that **B′** and **D′** are located on the other side of the window from **A** and **C**). As propylene is not symmetrical about the central carbon, it has two distinct adsorption sites: the high-energy configuration **A** at $z = 0.14$, in which the methyl group is located near the window and the low-energy configuration **B′** at $z = 0.43$, in which the smaller, unsaturated methylene is located near the window.

For propylene, configuration **B′**, where the C = C double bond is adjacent to the window, is 17.1 kJ/mol energetically more favourable than configuration **A** because the attractive guest–host interactions (π–π interactions and five C–H⋯F contacts shown in Fig. 2) are stronger (by 12.9 kJ/mol) while guest (by 0.3 kJ/mol) and host (by 3.9 kJ/mol, associated with smaller changes in pyrazine tilts, Supplementary Fig. 6) strain energies are lower (Supplementary Table 5). In configuration **A**, the C = C bond is further from the window, removing the π–π interactions (because the methylene, rather than methyl, terminus approaches the polyatomic anion), and only one C–H⋯F contact is formed with the $(NbOF_5)^{2-}$ anion from the methylene of the propylene (Fig. 2).

For propane, there are two broad adsorption sites **C** and **D′** equidistant from the window. Similar to propylene in configuration **A**, in both **C** and **D′** a methyl group is located near the window but sees different orientations of anions (Fig. 2), resulting in a small energy difference between **C** and **D′**.

The dynamical flexibility of the window defined by the four pyrazine molecules (Supplementary Fig. 6) enables guest diffusion. Simultaneous co-operative modulation of the pyrazine tilts (by up to 9.4° for propylene and up to 15.3° for propane) opens the window during the passage of the guest (Supplementary Fig. 6). The widest window diameter along the MEP was larger for propane (3.07 Å) than propylene (2.93 Å), with the enhanced tilt modulation to achieve this contributing to the larger barrier for propane. When either propane or propylene passes through the window, the anion orientations remain similar to those in the empty structure (Supplementary Fig. 7), maintaining the anion-pyrazine C–H⋯F contacts.

To build a complete picture of guest and host dynamics, in particular, the role of the anions that define the constriction, we conducted ab initio MD simulations at 300 K starting from configurations **A** and **C** and monitored guest position and anion orientation over 16 ps. Supplementary Fig. 8 shows that, whereas propylene resides near the window defined by the pyrazines at the cavity end ($z = 0.25$) consistent with the global minimum **B′** identified in the MEP calculations, the situation for propane is different. Propane does not remain at site **C**: it oscillates around the centre of the cavity ($z = 0$) near the constriction defined by the anions, accessing this position by inducing a rotation of one of the four $(NbOF_5)^{2-}$ anions that persists with time (Supplementary Fig. 8). An understanding of the MEP throughout the

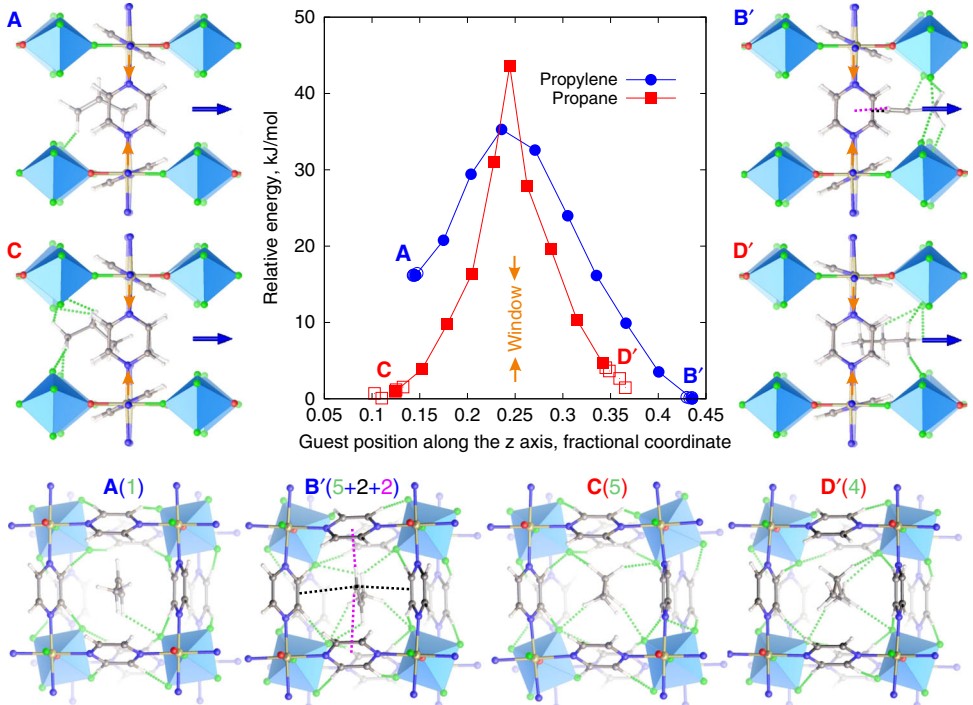

**Fig. 2 DFT calculations of guests near the window.** The local adsorption minima (empty symbols) for propylene (five overlapping blue circles for each site) and propane (five overlapping red squares for each site) near the window (marked by the orange arrows in the snapshots and in the graph at $z = 0.25$; the anions are located at $z = 0$ and $z = 0.5$) and the representative minimum energy path (MEP) profile calculated using the climbing image Nudged Elastic Band (NEB) DFT method[14] (filled symbols) for propylene (blue circles) and propane (red squares) in **NbOFFIVE**-1-Ni. The energy relative to the lowest energy configuration is shown as a function of the position of the central carbon on the $C_3H_6$ or $C_3H_8$ guest molecule. The snapshots show the stable configurations for propylene (**A** and **B′**) and propane (**C** and **D′**) on each side of the window. The guests rotate by 90° as they pass through the window owing to the shape of the channel. The pyrazine rotation as a function of guest position for both guests is shown in Supplementary Fig. 6. The green dotted lines show the C–H⋯F contacts shorter than 2.67 Å, the sum of van der Waals radii for H and F, with their total number indicated in brackets. Face-to-face (black dotted lines showing two carbon-to-double-bond distances of 3.18 Å each) and edge-to-face (magenta dotted line showing hydrogen-to-centroid distances of 2.75 Å and 3.35 Å) π-π interactions are shown for propylene in **B′**.

entire cell beyond the region near the window is therefore needed for both guests.

Figure 3a shows the MEP for both guests passing the constriction at $z = 0$ in the middle of the cavity, thus mapping the complete path of the guests through the unit cell. The MEP for propylene has an energy barrier (at $z = 0.03$) as propylene passes the anion, while its minima remain at the cavity end adjacent to the window. In strong contrast, there is a new adsorption site **E** for propane in the cavity centre ($z = 0$) near the anion that is 8.4 kJ/mol lower in energy than site **C** at the cavity end. Configuration **E** is the global minimum location for propane in **NbOFFIVE**-1-Ni. The unique feature of **E** is that it is created by a 31° rotation of one of the anions from its equilibrium orientation while the other anions maintain the orientation observed in an empty cell. This anion reorientation is essential to create the space required in order for propane to occupy this part of the cavity (Fig. 3b). The anion reorientation is driven by attractive interactions between propane and $(NbOF_5)^{2−}$, forming a pair of C–H⋯F contacts (2.37 Å and 2.62 Å) between the central $CH_2$ unit of the propane and the two fluorides on the anion and five C–H⋯F contacts under 2.67 Å to the other anions forming the cavity—the $CH_2$ unit participates in four contacts and the two methyl groups in three. Thus, configuration **E** is more energetically favourable than configuration **C**, where only five C–H⋯F interactions are formed rather than seven because one of the methyl groups is located too far away from the equatorial F atoms and the central $CH_2$ unit does not form four contacts created by anion rotation. As a result, configuration **E** produces

stronger attractive guest–host interactions by 13.0 kJ/mol that outweigh its higher host (by 3.9 kJ/mol) and guest (by 0.7 kJ/mol) strain energies (Supplementary Table 6).

**Structural experimental data.** To verify the guest location and the structural changes in the occupied host predicted with DFT, single-crystal X-ray diffraction experiments were performed on two **NbOFFIVE**-1-Ni crystals loaded with propylene (2 bar, 298 K) and propane (7 bar, 260 K), respectively. The different conditions were chosen to provide a comparable loading of both guests. In both cases, the availability of the computed models facilitated interpretation of the electron density associated with the guest. The introduction of propylene to the empty **NbOFFIVE**-1-Ni crystal structure results in a volume expansion of 2.1% (Supplementary Tables 1 and 2) of the $P4/nbm$ unit cell. The difference Fourier map reveals diffuse electron density with two pairs of peaks located near the window defined by the pyrazines on the fourfold rotation-inversion axis (Fig. 4a). The propylene location in the calculated lowest energy structure **B** suggests that peaks Q1 and Q2 correspond respectively to terminal $sp^3$- and $sp^2$-hybridised carbon atoms of a single disordered propylene molecule located in each cavity at $z = 0.439(1)$, leading to refinement of the experimental structure with the propylene molecule, located on the plane $m$ normal to one of the pyrazines, disordered over four equivalent positions at minimum **B** (Supplementary Fig. 10). The refined occupancy of 0.846(8) corresponds to one guest per cavity with ~85% cavities occupied and

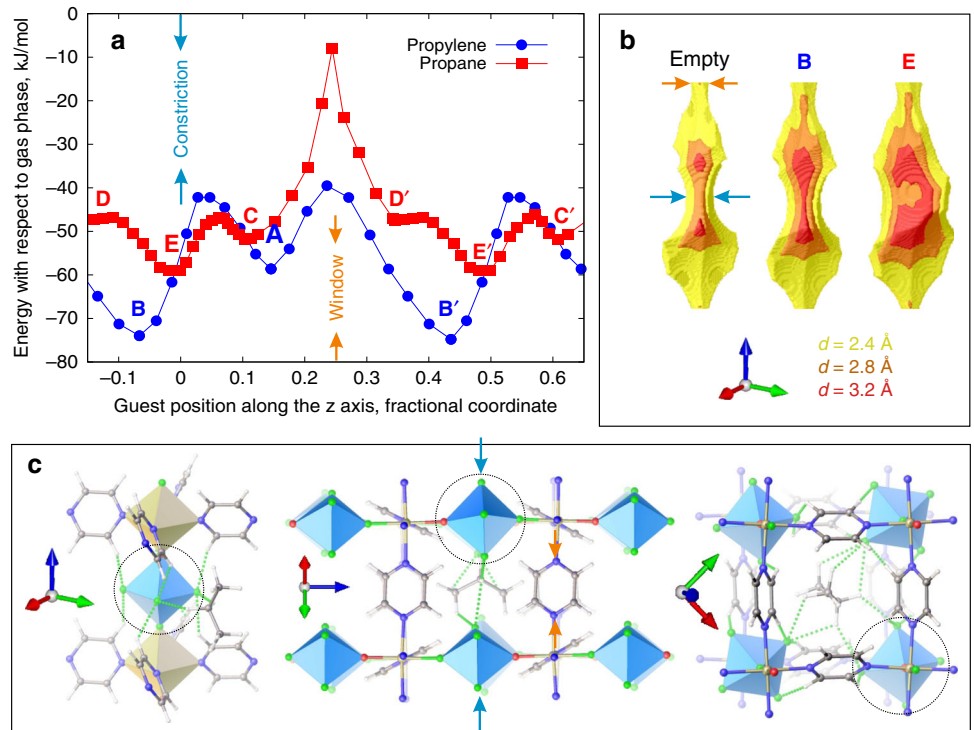

**Fig. 3 DFT calculations of the complete MEP through the unit cell. a** The representative diffusion profiles calculated with climbing image NEB DFT for propylene (blue circles) and propane (red squares) in **NbOFFIVE**-1-Ni. The guest energy relative to the gas phase is shown as a function of the position of the central carbon on the $C_3H_6$ or $C_3H_8$ molecule. **b** The shape of the cavity depicted using three probe sizes with diameters 2.4 Å, yellow, 2.8 Å, orange, 3.2 Å, red, for empty host and in the global minimum configurations **B** and **E**. Compared with the empty host, the cavity changes little in the presence of propylene (minimum **B**) but considerably in the presence of propane (minimum **E**). **c** Three views of the global energy minimum **E** in which the propane guest is located in the centre of the cavity ($z = 0$) with the adjacent $(NbOF_5)^{2-}$ anion rotated around the c axis (c.f., Fig. 1c) to form two C–H···F contacts with the central carbon. The anion rotation changes the pattern of the intra-framework C–H···F contacts, as now equivalent pyrazines form C–H···F contacts with the same equatorial fluoride (c.f., Fig. 1c). The views along the channel and through the window show seven attractive C–H···F contacts in **E** that lead propane to produce a distortion of the framework by anion rotation (circled) rather than adopt metastable minimum **C** where only five such contacts are possible (c.f., Fig. 2). Owing to the interactions with the anions, the propane in minimum **E** in the centre of the cavity is oriented diagonally across the cavity in contrast to all guests in Fig. 2 whose orientation is defined by the excluded volume interactions with the pyrazines. The guest orientation at 300 K as a function of its location in the cavity is shown in Supplementary Fig. 9.

compares well with the 76% observed in the sorption experiments at 2 bar and 298 K. Depending on the orientation of the adjacent anions (Supplementary Fig. 11), each propylene position near the pore window is stabilised by four close C–H···F contacts in the range of 2.47–2.65 Å (Supplementary Fig. 12a), π–π and C–H···π interactions between the propylene C = C bond and four adjacent pyrazine rings (Supplementary Fig. 12b), with the distance between the terminal methylene C of the guest and centre of the adjacent C–C bond of the pyrazine of 3.18 Å (face-to-face π–π interactions, exactly matching those identified computationally in Fig. 2), and between the H-atom of the terminal methylene group of the guest and the centroid of the two other pyrazines of 3.25 and 3.30 Å (C–H···π interactions, similar to those identified computationally in Fig. 2). The refined propylene position and orientation agree well with the calculated lowest energy configuration **B** (Fig. 4b).

Two sets of Ni-pyrazine layers are observed in the crystal structure: 85% pyrazines have a tilt of +23.9(2)°, whereas 15% have a tilt of −16.3(7)° that clashes with the location of the guest molecule and therefore originates from the cavity end not occupied by a guest (Supplementary Fig. 13). In contrast to the desolvated structure, there are four orientations of $(NbOF_5)^{2-}$ anions: 64(2)% at ±10.3(2)° and 36(2)% at ±34.2(6)°, respectively (Supplementary Fig. 11). The majority disorder component with +23.9(2)° pyrazine tilt and ±10.3(2)° anion orientation agrees well with the calculated lowest energy configuration **B** (Fig. 4b).

The introduction of propane to the empty **NbOFFIVE**-1-Ni crystal structure results in a volume expansion of 3.7% (1% along *a* and *b*, and 1.7% along *c*, see Supplementary Tables 1 and 3) and disorder between pyrazine tilts in the adjacent layers which changes the symmetry to $P4/mmm$. In the propane-loaded sample significant diffuse electron density was observed in the difference Fourier map near the constriction at the cavity centre (Fig. 4c), contrasting with the observation of electron density near the window for propylene and consistent with the computational prediction of different locations for propane and propylene. Propane is located at $z = 0$ close to the intersection of the fourfold axis with five mirror planes, generating disorder over four positions (Supplementary Fig. 14a) with total occupancy 0.808(6). The location and orientation of propane in the centre of the cavity (Supplementary Fig. 14b, c) causes 60% of $(NbOF_5)^{2-}$ anions to rotate and adopt $\varphi = ±36(1)°$ (Supplementary Fig. 15), whereas 40% of $(NbOF_5)^{2-}$ anions unaffected by propane stay close to the orientation observed in the empty structure ($\varphi = ±10$ (1)°). On the basis of the computed minimum **E** for a single propane in a single pore, with each propane at random inducing a rotation of one of the four neighbouring anions, 61% of the anions are expected to rotate (Supplementary Fig. 17). Each propane position is stabilised by up to five C–H···F contacts in the range of 2.45–2.52 Å (depending on adjacent anion orientations, Supplementary Table 4, Supplementary Fig. 16) to all three carbons of the propane molecule. The guest position and

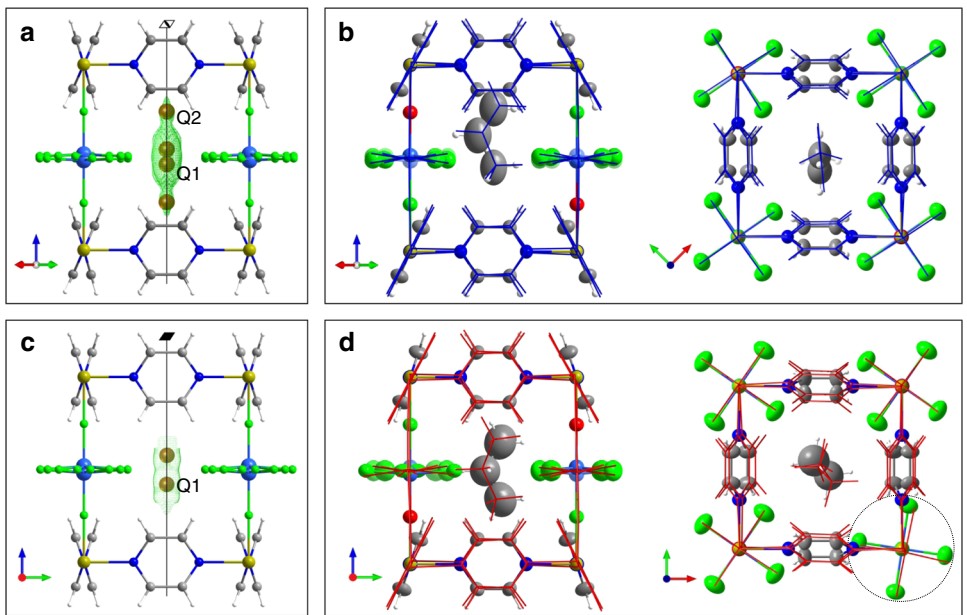

**Fig. 4 Crystal structures of NbOFFIVE-1-Ni with guests. a** Difference Fourier map for **NbOFFIVE**-1-Ni·0.85(propylene) showing diffuse electron density and two pairs of peaks (Q1 and Q2) in the cavity corresponding to a disordered propylene molecule. **b** Overlay of the experimental **NbOFFIVE**-1-Ni·0.85 (propylene) (displacement ellipsoids at 50% probability) and DFT (blue sticks corresponding to the global energy minimum **B**) models of a single cavity. **c** Difference Fourier map for **NbOFFIVE**-1-Ni·0.81(propane) showing diffuse electron density and a pair of equivalent peaks (Q1) in the cavity corresponding to a disordered propane molecule. **d** Overlay of the experimental **NbOFFIVE**-1-Ni·0.81(propane) (displacement ellipsoids at 50% probability, the rotated anion is circled) and DFT (red sticks corresponding to the global energy minimum **E**). **b** and **d** show one of the four orientations of the guest with the associated framework components, from the directions perpendicular and along the channel.

orientation of both dynamic framework units, anions and pyrazines, are in excellent agreement with calculated minimum **E**.

## Discussion

Computation and experiment show that propylene has a simple trajectory through **NbOFFIVE**-1-Ni. It preferentially resides at the end of the cavity near the window because of π–π interactions between its sp$^2$ carbons and the aromatic ring of pyrazine. Transport from one cavity to another requires co-operative rotation of the four pyrazines defining the window. The anion introduces a second barrier because it defines the constriction in the cavity and needs to rotate as the propylene passes.

The energy landscape and associated trajectory is more complex for the larger saturated propane guest, because, whereas it needs to trigger even larger co-operative pyrazine rotations to move along the channel, unlike propylene it does not reside near the window. Instead, propane in **NbOFFIVE**-1-Ni is located away from the window at the centre of the cavity, where there is originally a constriction in the undistorted **NbOFFIVE**-1-Ni host. Rotation of the $(NbOF_5)^{2-}$ anion is required to open the space that allows propane to occupy this position and optimises C–H···F interactions to the central methylene of the guest, as the sp$^3$ arrangement of the two C–H bonds matches the F···F separation of the equatorial fluorides of the rotated anion, whereas the terminal methyl groups interact with fluorides on the three other anions defining the constriction. This anion rotation is observed experimentally and can be shown computationally to be present at higher loadings regardless of whether the neighbouring cavities are occupied by propane or, as in the case of co-adsorption, propylene molecules (Supplementary Fig. 18).

The resulting location of propane in the cavity centre near the original constriction slows its passage through the channel by reducing the frequency of attempts to cross the window, whereas propylene is located next to the window and able to pass through it when the co-operative pyrazine rotation occurs. Owing to its

larger size, propane also requires a larger window opening than propylene, resulting in a higher diffusion barrier for propane. As the slower diffusion expected for propane is reinforced by the thermodynamic preference for propylene uptake, propylene is preferentially adsorbed by **NbOFFIVE**-1-Ni.

To verify this proposed sieving mechanism, we repeated NEB calculations for propylene and propane diffusion pathways in **SIFSIX**-3-Ni for which no propane exclusion is observed experimentally even at low pressures (Supplementary Fig. 3b). Figure 5 shows the MEPs for both guests in **SIFSIX**-3-Ni and the change in the shape of the pore cavity they produce. While there is still a marked difference between the calculated diffusion barrier of propane and propylene in **SIFSIX**-3-Ni (15.4 kJ/mol, compared with 15.9 kJ/mol in **NbOFFIVE**-1-Ni), there is no induced host distortion caused by either guest, in contrast to **NbOFFIVE**-1-Ni. Both guests reside near the window in **SIFSIX**-3-Ni (configurations **B** for propylene and equivalent minima **C** and **D** for propane in Fig. 5) and do not trigger anion rotation. There is a metastable state **E** for propane in the centre of the cavity that, unlike **NbOFFIVE**-1-Ni, does not produce anion rotation and is not the global minimum. The energy of state **E** in **SIFSIX**-3-Ni is 3.2 kJ/mol higher than global minimum **C**–the attractive interactions comprised of non-directional van der Waals interactions and in each case four C–H···F contacts (Fig. 5c) are stronger in **C** by 4.5 kJ/mol (78.1 vs 73.6 kJ/mol) and produce slightly higher framework strain energy (11.9 vs 10.1 kJ/mol) but slightly lower guest strain energy (0.7 vs 1.3 kJ/mol). In contrast, in **NbOFFIVE**-1-Ni, the attractive interactions at **E** of 83.7 kJ/mol include seven C–H···F contacts and host (21.9 kJ/mol) and guest (2.5 kJ/mol) strain energies are higher.

In both materials the separations of the centres of the polyatomic anions are defined by the same Ni-pyrazine-Ni distances. The smaller size of the $(SiF_6)^{2-}$ anion compared with $(NbOF_5)^{2-}$ is the main reason why propane does not drive anion rotation in **SIFSIX**-3-Ni, as the equatorial fluorides do not extend as far into

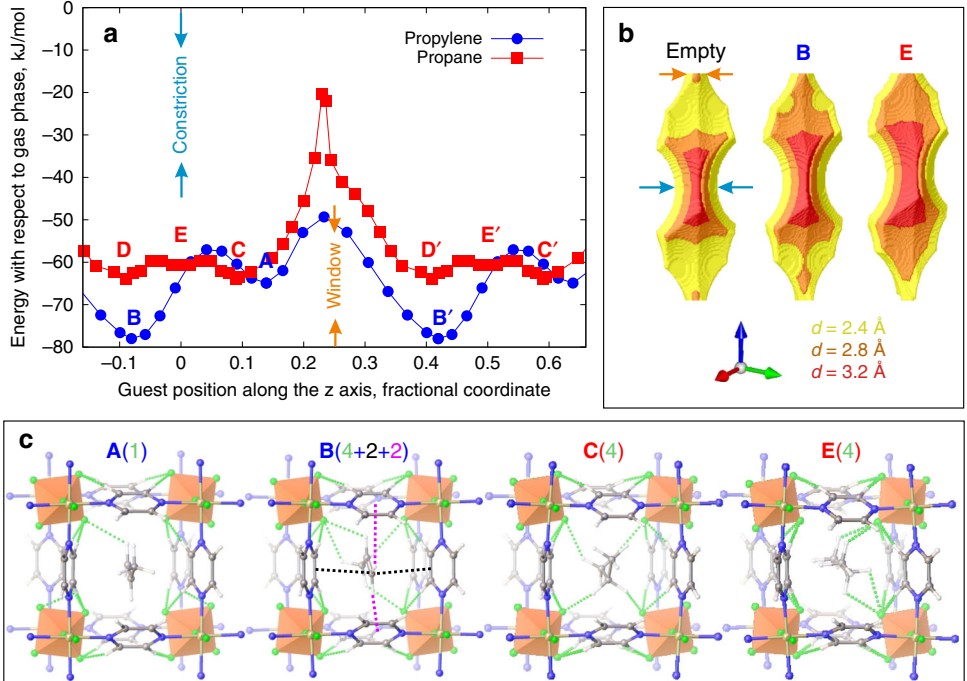

**Fig. 5 DFT calculations for SIFSIX-3-Ni. a** Complete MEPs calculated using the climbing image NEB DFT method for propylene (circles) and propane (squares). The energy relative to the lowest energy configuration is shown as a function of the position of the central carbon on the $C_3H_6$ or $C_3H_8$ guest molecule. The position of the Ni-pyrazine layer at $z = 0.25$ separating two adjacent cavities (centred at $z = 0$ and $z = 0.5$) is noted by orange arrows. **b** The shape of the cavity of an empty **SIFSIX**-3-Ni host and configurations **B** and **E**. Unlike in **NbOFFIVE**-1-Ni, the local energy minimum **E** is metastable and does not show any significant anion rotations. Similar to propylene, in **SIFSIX**-3-Ni propane preferentially resides near the window in one of the equivalent sites **C** or **D**, in contrast to the behaviour in **NbOFFIVE**-1-Ni. **c**, the snapshots show the stable configurations for propylene (**A** and **B**) and propane (**C** and **E**) with the total number of the C–H⋯F contacts shorter than 2.67 Å indicated in brackets. π–π (black dotted lines) and C–H⋯π (magenta dotted lines) interactions are shown for propylene.

the cavity. The constriction is thus wider, the anion does not need to rotate to accommodate propane, and is unable to form as many C-H⋯F contacts because of the greater distance to the cavity centre. The C–H⋯F contacts shorter than 2.67 Å are only formed with two out of four anions at a time in **SIFSIX**-3-Ni, unlike **NbOFFIVE**-1-Ni where all four anions can simultaneously form one or two C–H⋯F contacts with the propane guest, thus stabilising its position in the centre of the cavity through octahedral rotation. The wider cavity constriction produced by the smaller $(SiF_6)^{2-}$ anions means that in **SIFSIX**-3-Ni, unlike in **NbOFF-IVE**-1-Ni, propane is stabilised near the window, similarly to propylene. Based on the Boltzmann populations at 298 K, state **E** is 13.2 times more likely to be occupied than **C** in **NbOFFIVE**-1-Ni, but 3.5 times less likely to be populated than **C** in **SIFSIX**-3-Ni. Thus **NbOFFIVE**-1-Ni reduces the crossing attempt frequency for propane by holding it in the centre of the cavity, whereas **SIFSIX**-3-Ni encourages propane diffusion by placing it near the window.

The cavity width and length in **NbOFFIVE**-1-Ni are set by the pyrazine and the $(NbOF_5)^{2-}$ anion, respectively. The resulting dimensions match propane and propylene so closely that only one guest can occupy a single cavity at a time. The synergy between anion control of propane location within the cavity and pyrazine control of cavity-to-cavity hopping couples guest with host dynamics to differentiate propane and propylene transport and uptake. The appropriately matched host–guest dynamics harness the flexibility of the whole pore system, not solely the window, to optimise separation. This contrasts with the separation mechanism in larger pore flexible materials such as ZIF-8[15–17], where both propane and propylene can occupy positions near and far away from the windows, and the pore surface chemistry does not

discriminate them through multiple supramolecular interactions. This suggests that MOF with multiple linkers that distinguish similar guests by defining separate positions for them in the pore space through differentiated dynamical response are promising candidates for challenging separations.

## Methods

**High-pressure gas adsorption studies**. High-pressure gas adsorption studies were performed on a magnetic suspension balance marketed by Rubotherm (Germany). Type Adsorption equilibrium measurements of pure gases were performed using a Rubotherm gravimetric-densimetric apparatus G-Hp-Flow, composed mainly of a magnetic suspension balance and a network of valves, mass flow metres, and temperature and pressure sensors (Supplementary Note 1). Prior to each adsorption experiment, ~100 mg of the sample was outgassed at 378 K for 12 h under a residual pressure of $10^{-6}$ mbar. The temperature during adsorption measurements is maintained constant using a thermostat-controlled circulating fluid. To study co-adsorption of propane and propylene on **NbOFFIVE**-1-Ni, the Rubotherm gravimetric system was coupled with Dani GC. After outgassing the sample, a fixed amount of premixed $C_3H_6/C_3H_8$:50/50 mixture was introduced to the system. After equilibrium was reached in 17 h at 1.3 bar and 298 K, the composition of non-adsorbed phase was analysed using GC (Supplementary Note 2).

**Single-crystal X-ray diffraction**. The as-made **NbOFFIVE**-1-Ni crystals were placed into the gas cell and then activated at 105 °C under $N_2$ flow for 24–48 h and then in vacuum for 6 h while heating. After activation, gas was introduced slowly into manifold at the pressure just above 1 bar. The pressure was incrementally increased to 2 bar in case of propylene, whereas 7 bar in case of propane. After the equilibration time of 1 hour in the manifold, the gas cell was removed while maintaining the pressure inside capillary.

Single-crystal X-ray diffraction data were collected using a Bruker X8 PROSPECTOR APEX2 CCD diffractometer equipped with Cu Kα tube ($\lambda$ = 1.54178 Å). Indexing was performed using APEX2 (Difference Vectors method)[18]. Data integration and reduction were performed using SaintPlus 8.34A[19]. Absorption correction was performed by analytical method implemented in SADABS[20]. Space groups were determined using XPREP implemented in APEX2. Structures were solved using SHELXS-97 (direct methods) and refined using

SHELXL-2014 (full-matrix least-squares on $F^2$) contained in WinGX[21,22]. Crystal data and refinement conditions are shown in Supplementary Tables 1–3.

**DFT calculations**. We use DFT to assess the role of host–guest interactions by optimising the structure of a fully flexible unit cell of **NbOFFIVE**-1-Ni or **SIFSIX**-3-Ni containing four formula units that form two parallel channels each comprising of two consecutive pore cavities with and without a propane or propylene molecule adsorbed in one of the cavities. All periodic DFT calculations were carried out using Vienna Ab Initio Simulation Package (VASP) version 5.3.5. To accurately describe dispersion interactions we used optB86b-vdW exchange functional[23] in the framework of the generalised gradient approximation (GGA)[24] using the projector augmented wave (PAW)[25] pseudopotentials ver. 54 with the energy cutoff of 520 eV.

We use NEB calculations[26] to capture the energy profile and structural changes associated with guest transport between the known energy minimum configurations. The NEB calculations with recommended default VASP settings involved 6–12 system replicas and used simulation parameters identical to those in our energy minimisation calculations.

We also conduct ab initio MD to model the system's dynamics at 300 K (simulation time 16 ps, time step 0.5 fs) and assess host dynamics manifested by the change in the orientation of two structural elements: pyrazine molecules and polyatomic anions, and how the guests affect them.

The full list of simulation parameters and the formulas used to calculate binding and strain energies are given in Supplementary Note 3.

## Data availability

The X-ray crystallographic coordinates for structures reported in this study have been deposited at the Cambridge Crystallographic Data Centre (CCDC), under deposition numbers 1997482–1997484. The simulation data that support the findings of this study have been deposited in DataCat repository: http://datacat.liverpool.ac.uk/1083 (https://doi.org/10.17638/datacat.liverpool.ac.uk/1083).

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

## Acknowledgements

This publication is based upon work supported by the King Abdullah University of Science and Technology (KAUST) Office of Sponsored Research (OSR) under award number OSR-2016-RPP-3273. We acknowledge the HPC Materials Chemistry Consortium for providing access to UK's national supercomputer ARCHER under the EPSRC grant EP/L000202 and CCF project as well as KAUST Supercomputing Laboratory for providing access to supercomputer Shaheen. D.A. thanks the Leverhulme Trust for funding via the Leverhulme Research Centre for Functional Materials Design (RC-2015-036). M.J.R. is a Royal Society Research Professor.

## Author contributions

D.A., M.S.D. and M.J.R. conceived the theoretical part of the project. D.A. and M.S. performed calculations. Y.B. and M.E. designed the experimental approach. A.C. and K.A. synthesised and characterised the samples, P.M.B. performed adsorption experiments, A.S. carried out the SCXRD experiments and interpreted structural data. D.A., A.S., M.S.D., M.E. and M.J.R. wrote the paper. All authors discussed the results and implications and commented on the manuscript.

## Competing interests

The authors declare no competing interests.
