## [Peer Review File · Nature Communications]

REVIEWER COMMENTS

Reviewer #1 (Remarks to the Author):

Propylene/propane separation is of paramount importance, and the titled MOF NbOFFIVE-1 has been reported to show the very useful molecular sieving separation under ambient condition, while the isostructural MOF SIFSIX-3-Ni does not exclude propane at the same condition. This work uses computer simulation, crystallography, and adsorption data to reveal the mechanisms, which is potentially very important in this field. The results and conclusions, associated with the structural flexibility of the MOF, are reasonable and suitable for explaining the very different adsorption behaviors for propylene and propane. However, there are some issues needed to be considered. Since both guests can induce expansion of the framework, the authors should consider the co-adsorption of propane after adsorption of propylene. Although the crystal structures showed that propane can induce a larger expansion, this does not mean that the slightly smaller framework cannot adsorb small amount of propane, because adsorption and structure distortion can occur non-periodically, and the crystal structure is an average result. Further, as shown in supplementary Fig. 4, the equilibrium propane adsorption under ambient condition should be significant. This implies that if the framework is opened by propylene, propane may also be able to enter the pore. Finally, a recent paper (Angew. Chem. 2019, 58, 7692) demonstrated adsorption of propane in the propylene/propane mixture.

The crystal data, such as the CIF files, should be submitted for evaluation.

Reviewer #2 (Remarks to the Author):

This is a very elegant computational/experimental study which clearly shows the potential of a MOF system to provide effective separation of propylene from propane - a key process in the chemicals industry; an interesting contrast is presented with a second MOF system. Computation and experiment are very effectively combined to provide a molecular level understanding of the basis of the observations. More generally, the paper provides a beautiful illustration of molecular sieving by microporous materials. The work is of the quality and general interest to justify publication in Nature Comms.

The authors should add some clarification of the DFT techniques in the main body of the text and not just in the SI. In particular, the fact that 4 unit cells were used in the repeat unit and the value of time step for the MD are important parameters which need to be in the main text.

A more general consideration for the future is that it should be quite possible to derive an effective force field for these systems which would allow much longer MD simulations, hence giving richer information on the dynamics.

Richard Catlow

POINT-BY-POINT RESPONSE TO REVIEWER COMMENTS

Reviewer #1 (Remarks to the Author):

Comment 1. *Propylene/propane separation is of paramount importance, and the titled MOF NbOFFIVE-1 has been reported to show the very useful molecular sieving separation under ambient condition, while the isostructural MOF SIFSIX-3-Ni does not exclude propane at the same condition. This work uses computer simulation, crystallography, and adsorption data to reveal the mechanisms, which is potentially very important in this field. The results and conclusions, associated with the structural flexibility of the MOF, are reasonable and suitable for explaining the very different adsorption behaviors for propylene and propane.*

Response 1. We thank Reviewer 1 for encouraging comments.

Comment 2. *Since both guests can induce expansion of the framework, the authors should consider the co-adsorption of propane after adsorption of propylene. Although the crystal structures showed that propane can induce a larger expansion, this does not mean that the slightly smaller framework cannot adsorb small amount of propane, because adsorption and structure distortion can occur non-periodically, and the crystal structure is an average result. Further, as shown in supplementary Fig. 4, the equilibrium propane adsorption under ambient condition should be significant. This implies that if the framework is opened by propylene, propane may also be able to enter the pore. Finally, a recent paper (Angew. Chem. 2019, 58, 7692) demonstrated adsorption of propane in the propylene/propane mixture.*

Response 2. We thank Reviewer 1 for raising this pertinent point and for bringing this publication to our attention. We added a reference to this interesting and relevant work on propylene/propane separation. Since the framework volume changes by only 2.1% when 80% of pores are filled with propylene, we argue that the framework NbOFFIVE-1-Ni is different from other flexible MOFs reported in the literature. The NbOFFIVE-1-Ni MOF does not undergo significant structural expansion/contraction on guest adsorption/removal as observed in flexible MOFs; rather NbOFFIVE-1-Ni has components with some rotational freedom, pyrazines and anions which respectively result in dynamic window and dynamic cavity of the channel. Modelling shows that the dynamic window transiently opens to allow propylene to enter the channel and then closes again. The same process is repeated for a second molecule entering the channel. In simple words, the window acts like a revolving door, allowing one molecule at a time, hence it is not plausible that propane will enter the pore through the window already opened by propylene. This is demonstrated by the calculated pyrazine orientations in Supplementary Fig. 6: when a propene molecule resides near its equilibrium position, the pyrazines adopt their original orientation and the window is closed.

We agree with the Reviewer's comment that some co-adsorption of propane with propylene is possible. However, the reported amounts are well under 10% even at high pressures of 8 bar in our adsorption experiments and of 15 bar in *Angew. Chem. 2019, 58, 7692*. Regardless the very small amount of propane co-adsorbed with propylene, the key to the complete separation of propane from propylene in breakthrough experiments with NbOFFIVE-1-Ni is the complete uptake of propylene by the bed as originally presented in *Science* **2016**, 353 (6295), 137 and briefly reviewed below with the focus on the relevant co-adsorption data.

Figure 1 shows breakthrough experiment for a C₃H₆/C₃H₈ 50/50 gas mixture at 298 K that demonstrates complete separation between propane and propylene, with no propylene coming out of the column for first 7 minutes. The analysis of the adsorbed

phase after this breakthrough experiment confirms the predominance of propylene adsorption as the experiment shows only a weak signal corresponding to propane (Figure 2). Additionally, a further breakthrough experiment for a ternary mixture $C_3H_6/C_3H_8/N_2$: 25/25/50, demonstrates simultaneous breakthrough of propane and N_2 confirming that propane behaves as N_2 which is only weakly absorbed by the material (Figure 3).

Finally, as shown in Supplementary Figure 4a, there is practically no propane adsorption below 2 bar even in the absence of propylene. The adsorption of propane above 2 bar has very slow kinetics (Figure 4) and is not expected to give propylene much competition for adsorption even at higher pressure.

To explicitly address the point about co-adsorption raised by Reviewer 1, we have updated the following sentence:

The total gravimetric adsorption uptake of the equimolar propylene/propane gas mixture at pressures up to 8 bar closely matches the uptake of pure propylene at the same partial pressure of propylene (Supplementary Fig. 5) at both 25°C and 50°C, indicating that the amount of propane co-adsorbed with propylene in NbOFFIVE-1-Ni remains very low even at high pressure and temperature.

Comment 3. The crystal data, such as the CIF files, should be submitted for evaluation.

Response 3. We have included CIF files with revised version and deposited the files to CCDC, deposition numbers 1997482-1997484.

Figure 1. 1st cycle in the series of repetitive column adsorption breakthrough test using C_3H_6/C_3H_8 : 50/50 at 298 K, 1 bar and using 4 cm³/min.

Figure 2. Mass spectrometric analysis of the bed downstream during the desorption process (after passing C_3H_6/C_3H_8 : 50/50 mixed gas through the bed). Desorption was carried out by purging He ($40 \text{ cm}^3/\text{min}$) at 298 K. To remove unabsorbed gases from the line, the column was purged for 1 min before monitoring it with the mass spectrometer. As observed, the propylene signal is dominating over 10 minutes.

Figure 3. $C_3H_6/C_3H_8/N_2$: 25/25/50 mixed gas experiment using packed column bed at 298 K and 1 bar total pressure ($8 \text{ cm}^3/\text{min}$ total flow) after activation at $105 \text{ }^\circ\text{C}$ under vacuum for 8 hours.

Figure 4. Adsorption kinetics of propane for some of the equilibrium points during high pressure propane adsorption on NbOFFIVE-1-Ni described in Supplementary Figure 4.

Reviewer #2 (Remarks to the Author):

Comment 1. *This is a very elegant computational/experimental study which clearly shows the potential of a MOF system to provide effective separation of propylene from propane - a key process in the chemicals industry; an interesting contrast is presented with a second MOF system. Computation and experiment are very effectively combined to provide a molecular level understanding of the basis of the observations. More generally, the paper provides a beautiful illustration of molecular sieving by microporous materials. The work is of the quality and general interest to justify publication in Nature Comms.*

Response 1. We thank Reviewer 2 for encouraging comments.

Comment 2. *The authors should add some clarification of the DFT techniques in the main body of the text and not just in the SI. In particular, the fact that 4 unit cells were used in the repeat unit and the value of time step for the MD are important parameters which need to be in the main text.*

We thank Reviewer 2 for raising this point. We added this missing information to the main text in the Methods section to make it readily accessible to the reader. The modified text is highlighted below:

We use density functional theory (DFT) to assess the role of host-guest interactions by optimising the structure of a fully flexible unit cell of **NbOFFIVE-1-Ni** or **SIFSIX-3-Ni** containing four formula units that form two parallel channels each comprising of two consecutive pore cavities with and without a propane or propylene molecule adsorbed in one of the cavities.+

We also conduct *ab initio* MD to model the system's dynamics at 300 K (simulation time 16 ps, time step 0.5 fs) and assess host dynamics manifested by the change in the orientation of two structural elements: pyrazine molecules and polyatomic anions, and how the guests affect them.+

Comment 3. *A more general consideration for the future is that it should be quite possible to derive an effective force field for these systems which would allow much longer MD simulations, hence giving richer information on the dynamics.*

We fully agree with the Reviewer. Studying the dynamics of a larger system over a nanosecond time scale is an interesting subject for future studies that require the development of a dedicated force field.

While writing the paper we tested UFF4MOF and some of its modifications and found that these forcefields produced large pyrazine tilts and effectively closed the pore window for guest transport. While determining model parameters that capture the correct framework structure and the dynamics of both pyrazine and anion is relatively simple, it might be difficult to accurately describe guest-host interactions such as pi-interactions for propene and the balance between weak C-H...F contacts and van der Waals interactions for propane. We were fortunate with the choice of the target system . a relatively small unit cell in which the guest can spontaneously move from one side of the pore cavity to the other on a picosecond time scale. Thus we were able to study guest dynamics within the pore cavity and its effect on the framework structure with *ab initio* MD.

REVIEWER COMMENTS

Reviewer #1 (Remarks to the Author):

It seems that the authors tried to weaken the discussion about the point of co-adsorption. In my opinion, co-adsorption, or absence of co-adsorption, is very important for molecular sieving and flexible MOFs. Actually, the figures shown in the responses cannot exclude co-adsorption. Figure 1 means C₃H₆ and C₃H₈ can be well separated, but cannot judge if C₃H₈ is adsorbed; Figure 2 shows obvious adsorption of C₃H₈ (integration of the two curves can give quantitative information, although it is not directly related to the gases adsorbed in the MOF); Figure 3 means C₃H₈ adsorbed weakly like N₂, but cannot compare the adsorption amounts; Figure 4 is single-component behaviors, not directly related with mixture adsorption. Figure S5 was used to support the claim about very low co-adsorption in the new sentences. However, C₃H₈ is just 5% heavier than C₃H₆, which can be hardly distinguished by the gravimetric measurement, even if all gas adsorbed in the C₃H₆/C₃H₈ mixture is C₃H₈.

About the mechanism study, additional simulations using C₃H₆/C₃H₈ mixture as guest may be useful.

POINT-BY-POINT RESPONSE TO REVIEWER COMMENTS

Reviewer #1 (Remarks to the Author):

Comment 1. *It seems that the authors tried to weaken the discussion about the point of co-adsorption. In my opinion, co-adsorption, or absence of co-adsorption, is very important for molecular sieving and flexible MOFs. Actually, the figures shown in the responses cannot exclude co-adsorption. Figure 1 means C₃H₆ and C₃H₈ can be well separated, but cannot judge if C₃H₈ is adsorbed; Figure 2 shows obvious adsorption of C₃H₈ (integration of the two curves can give quantitative information, although it is not directly related to the gases adsorbed in the MOF); Figure 3 means C₃H₈ adsorbed weakly like N₂, but cannot compare the adsorption amounts; Figure 4 is single-component behaviors, not directly related with mixture adsorption. Figure S5 was used to support the claim about very low co-adsorption in the new sentences. However, C₃H₈ is just 5% heavier than C₃H₆, which can be hardly distinguished by the gravimetric measurement, even if all gas adsorbed in the C₃H₆/C₃H₈ mixture is C₃H₈.*

Response 1. We thank reviewer for their comments. We agree with reviewer that some of the data, especially when viewed in isolation, does not conclusively rule out propane co-adsorption, as it is always difficult to rule out co-adsorption in trace amount. However, the combined evidence from all these data suggest that co-adsorption of propane is not significant and it is possible to recover propylene with high purity of around 99% or more, which is targeted purity for propylene for polypropylene production and other industrial applications.

To unambiguously demonstrate that the co-adsorption of propane is extremely low, we performed new experiments using the GC-coupled mixed-gas gravimetric adsorption system. A single adsorption measurement at 25 °C was performed using a fixed amount of premixed high-grade C₃H₆/C₃H₈:50/50 mixture. The adsorption was allowed to equilibrate for 17 hours till the equilibrium was reached at 1.3 bar and then non-adsorbed phase was analyzed using GC to determine its composition. The results are shown in Figure 1, which is now included as revised Supplementary Fig. 5.

The comparison between adsorption and GC data indicate that adsorbed phase contains more than 99% propylene. We updated the main text to say

In the competitive adsorption of equimolar propylene/propane mixture at 25°C the co-adsorption of propane is expected to be very low as the total uptake of the mixture closely matches the uptake of pure propylene up to 3 bar at the same partial pressure of propylene (Supplementary Fig. 5a) as if propane was not present in the mixture. The gas chromatography analysis of the mixture uptake at 1.3 bar (Supplementary Fig. 5b) confirms that over 99% of adsorbed mixture is propylene, while the amount of co-adsorbed propane is below the experimental error of 2%.

and provided a detailed explanation of the experimental setup and how the uptake and the experimental error were calculated in Supplementary Information.

Figure 1. High-pressure adsorption data and GC analysis of non-adsorbed phase in NbOFFIVE-1-Ni. (a) High-pressure adsorption isotherms at 25 °C for pure propylene (blue) and a C₃H₆/C₃H₈:50/50 mixture (orange) compared with a single GC-coupled mixed gas adsorption measurement for C₃H₆/C₃H₈:50/50 at 1.3 bar total pressure. The difference between the blue and the orange curve at any given partial pressure of propylene is indicative of the small amount of propane co-adsorbed from the mixture (b) GC analysis of the non-adsorbed phase after adsorption of C₃H₆/C₃H₈:50/50 on NbOFFIVE-1-Ni, showing that the composition of gas mixture has changed from 50/50 to 45.2/54.8 due to selective adsorption of propylene by the material.

Comment 2. *About the mechanism study, additional simulations using C₃H₆/C₃H₈ mixture as guest may be useful.*

Response 2. We thank Reviewer 1 for encouraging comments. To quantify the effect of co-adsorption we performed additional DFT calculations in which adsorption of a propane molecule was studied in the presence of different amount of propylene adsorbed in the neighboring cavities. Two representative cases were considered for a NbOFFIVE-1-Ni layer containing four cavities: a propane molecule adsorbed near a propylene molecule with two other cavities remaining unoccupied (Fig. 2a) and a propane molecule surrounded by three propylene molecules (Fig. 2c) – the highest occupancy case. These two simulations were compared to calculations for similar cells containing a propylene molecule instead of the propane (Figs 2b and 2d). The side-by-side comparison indicates that the mechanism of preferential guest location described in the paper is not affected by the presence of propylene molecules. In both Fig. 2a and Fig. 2c the propane guest is located near the anion (unlike all propylene guests that are adsorbed at the pore window). In this position, the propane molecule has orientation distinct from that of propylene molecules and causes anion rotation - the anion in the centre of the snapshots has a different orientation from other anions. To emphasise the fact that the anion re-orientation by propane is unaffected by the presence of other guests in the neighbouring pores and indeed is observed experimentally at propane loading of 0.81, we added a sentence in the main text:

This anion rotation is observed experimentally and can be shown computationally to be present at higher loadings regardless of whether the neighbouring cavities are occupied

by propane or, as in the case of co-adsorption, propylene molecules (Supplementary Fig. 18).

Figure 2 is now included in the Supplementary Information as Supplementary Fig. 18. Its caption reports the calculated difference in propylene vs propane binding energies in b) compared to a) and in d) compared to c), that we used to estimate the equilibrium composition of the adsorbed phase at different pore loadings at 298 K. These estimates are in good agreement with the experimental GC data.

Figure 2. Computational assessment of propane co-adsorption. Lowest energy configurations were identified for 2x1x1 supercells containing several guest molecules in the neighbouring cavities: a) one propane and one propylene molecule b) two propylene molecules c) one propane and three propylene molecule d) four propylene molecules. The anion in the centre of a) and c) has a different orientation compared to other anions in the system due to the interaction with the propane molecule that is located near it. In the absence of propane, all anions in b) and d) have the same orientation. By comparing total binding energies, it is possible to tell that adsorption of a propylene molecule instead of propane is energetically favourable in b) compared to a) by $\Delta E=13.4$ kJ/mol and in d) compared to c) by $\Delta E=8.0$ kJ/mol. If we approximate the propylene/propane ratio in the material by that of a two-state statistical model, $e^{\Delta E/kT}/1$, and use these two calculated differences in binding energies, we obtain an estimate for the equilibrium compositions of propylene and propane in the adsorbed phase at $T=298\text{K}$ as respectively 99.6/0.4 for the loading of 0.5 and 96.2/3.8 for full loading. While this Boltzmann population analysis is approximate, our simulations clearly indicate that (1) the propane is located opposite the anion and the anion rotation still takes place even when all neighbouring cavities are occupied by propylene, (2) regardless of the pore loading, there is a thermodynamic preference to adsorb propylene instead of propane and (3) this preference reduces as the loading increases which is expected to increase the trace of propane in the adsorbed phase as the uptake increases with pressure at a

given temperature. These results are in qualitative and quantitative agreement with the available experimental data.

REVIEWERS' COMMENTS

Reviewer #1 (Remarks to the Author):

The authors made great efforts to address my concerns, and I think this revision is acceptable.